# Cross-sectional evaluation of an asynchronous multiple mini-interview (MMI) in selection to health professions training programmes with 10 principles for fairness built-in

Alison Callwood  ,[1] Jenny Harris,[1] Lee Gillam,[2] Sarah Roberts,[1] Angela Kubacki,[3] Angelos Christidis,[2] Paul Alexander Tiffin[4]

[1]Faculty of Health and Medical Sciences, University of Surrey, Guildford, UK
[2]Faculty of Engineering and Physical Sciences, University of Surrey, Guildford, UK
[3]IMBE, St George's University, London, UK
[4]Department of Health Sciences, University of York, York, UK

**Correspondence to**
Dr Alison Callwood;
a.callwood@surrey.ac.uk

## ABSTRACT

**Objectives** We aimed to explore the psychometric properties of the first known online asynchronous multiple mini-interview (MMI) designed for fairness with subgroup analyses by key characteristics, usability and acceptability.

**Design** Cross-discipline multimethod evaluation.

**Setting** One UK University.

**Participants** Applicants to nursing, midwifery and paramedic science undergraduate programmes during 2021–2022.

**Primary, secondary outcome measures** Psychometric properties (internal consistency, construct validity, dimensionality) were assessed using Cronbach's alpha ($\alpha$), parallel analysis (PA), Schmid-Leiman transformation and ordinal confirmatory factor analysis (CFA). Usability and acceptability were evaluated using descriptive statistics and conventional content analysis.

**Methods** The system was configured in a seven question 4 min MMI. Applicants' videorecorded their answers which were later assessed by interviewers and scores summed. Applicants and interviewers completed online evaluation questionnaires.

**Results** Performance data from 712 applicants determined good-excellent reliability for the asynchronous MMI (mean $\alpha$ 0.72) with similar results across subgroups (gender, age, disability/support needs, UK/non-UK). PA and factor analysis results suggested there were seven factors relating to the MMI questions with an underlying general factor that explained the variance in observed candidate responses. A CFA testing a seven-factor hierarchical model showed an excellent fit to the data (Confirmatory Fit Index=0.99, Tucker Lewis Index=0.99, root mean square error (RMSE) =0.034). Applicants (n=210) viewed the flexibility, relaxed environment and cost savings advantageous. Interviewers (n=65) reported the system to be intuitive, flexible with >70% time saved compared with face-to-face interviews. Reduced personal communication was cited as the principal disadvantage.

**Conclusions** We found that the asynchronous MMI was reliable, time-efficient, fair and acceptable and building fairness in was lost-cost. These novel, insights are applicable across health professions selection internationally informing the future configuration of online

## STRENGTHS AND LIMITATIONS OF THIS STUDY

⇒ The underpinning theoretical approach aligned with an iterative process necessary to design a new technology to reduce bias.
⇒ The large sample enabled us to assess psychometric properties with subgroup analyses for the first time in this context.
⇒ The study provides perspectives from one large site; a necessary step to inform a planned international multisite evaluation.
⇒ The multimethod design provided insights necessary to embed fairness into online selection approaches in the absence of best practice guidance.

interviews to ensure workforces represent the societies they serve.

## INTRODUCTION

Ensuring equity, inclusivity and diversity in selection to health professions training programmes is recognised internationally as an ethical and practical imperative.[1 2] Globalisation and increased workforce pressures amplify this need.[3] Fulfilling our responsibility to ensure fair selection is complex due to unintended biases that are intrinsic to human assessment compounded by recent unprecedented change to online interviews in the absence of published evidence.[4–7]

Historically, health professions' selection has been mainly face-to-face using unstructured or structured approaches including panel interviews, group interviews, assessment centres and multiple mini-interviews (MMIs).[8] MMIs are a series of short, focused interactions with a number of different interviewers. The multiquestion format featuring structured scoring proforma with interviewers who have no prior knowledge of applicants, is

designed to mitigate the potential impact of interviewer bias.[9] MMIs have been shown to be a feasible, acceptable, valid and reliable candidate selection approach across health professions. None-the-less, as a face-to-face method, MMIs can be costly, resource intensive and influenced by unintended bias.[10]

Online interviews were a relatively uncommon occurrence in selection to health professions before the pandemic. Approaches included Skype-based MMIs and asynchronous MMIs.[11] Research outside the field of healthcare has shown asynchronous video interviews to be faster, cheaper and require less employee time, easing scheduling burden and allowing for more applicants to be screened.[12] This can potentially increase the number of applicants who would have otherwise not had the opportunity to be interviewed.

During the Covid-19 pandemic, it was vital to ensure the continuance of recruitment to health professions. This resulted in rapid adaption to using online interviews facilitated by videoconference technology.[13 14] Recent research suggests online interviews like MMIs are feasible and acceptable provided reliable high-speed internet connection is available. However, access to reliable Wi-Fi is not always possible.[11] In an asynchronous approach, applicants record their interviews at a convenient time and place, alleviating potential technical issues.

In live synchronous interviews, nuanced inconsistencies in, for example, tone and intonation, can arise in the way interviewers ask questions. Consistency of questioning across applicants is assured in asynchronous interviews through the use of prerecorded interview questions. Fairness is further ensured with the avoidance of non-adherence to set questions which can occur in live interviews when applicants and interviewers serendipitously find something in common and deviate to discuss this.[12 15]

'Fairness' in this article is conceptualised as the quality of treating people equally or in a way that is right or reasonable. That encapsulates perceived fairness by participants as well as that borne out in data. Consensus on the design of online interviews to optimise applicant accessibility and usability and mitigate potential unfairness issues for people across demographics, abilities and disabilities is not readily available.[15] To address this, we applied to Innovate UK, the UK's innovation agency (2020–2021) to build and evaluate what we believe was, the first (proof-of-concept) asynchronous videoconference facilitated interview and assessment system uniquely grounded in the MMI method (figure 1).

### Asynchronous MMI platform development summary

The asynchronous MMI is an on-demand videoconference interview where applicants log onto an online portal to complete their interview. The interview is a remote experience, where the system records applicants' short video responses to MMI questions in a timed process emulating the face-to-face MMI method. There is no synchronous, bidirectional communication and no interviewer to interact with; instead, questions are prerecorded by interviewers. Applicants' recorded responses are scored by an interview assessor at a convenient later date.

To inform the design, we undertook a rapid review of literature published between 2011 and 2021, guided by a five-stage process[16 17] to help us understand how video-based interviews compare with face to face in terms of implementation and fairness including what makes for an optimising experience (details provided in online supplemental appendix 1).

Indicative themes were elicited and assimilated into the following 10 key principles and the system was reconfigured accordingly:
- ▶ Recognise potential issues with stereotype threats and belonging uncertainty that may impact on candidates' performance and use language that supports the affirmation of values, for example, 'well done for getting this far'.
- ▶ Incorporate encouraging words/phrases into the interview dialogue, as well as any communications circulated to applicants (eg, 'good luck').

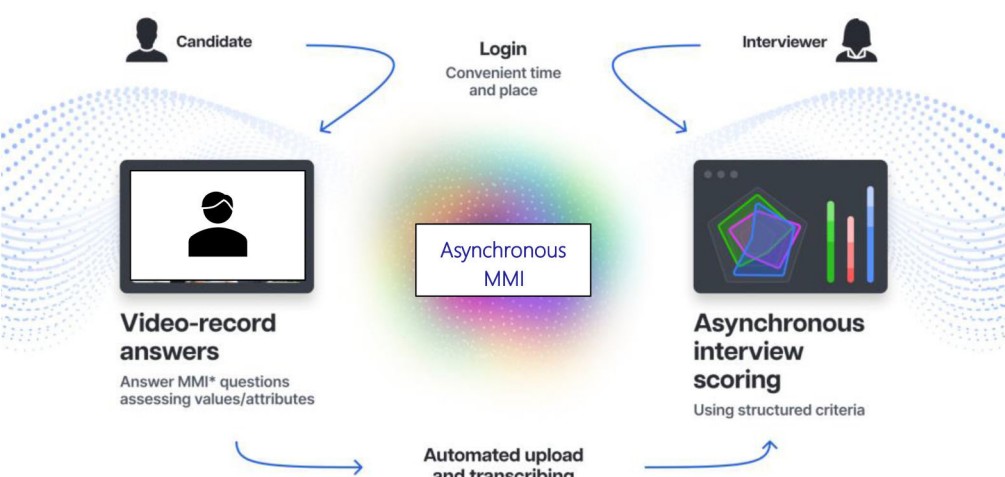

**Figure 1** Asynchronous Multiple Mini Interview (MMI) infographic.

- ► Soften the language of technical instructions, for example, 'when you are ready …' or 'when you have familiarised yourself with…'.
- ► Reduce the verbal load of interview content particularly for neurodivergent applicants.
- ► Accommodate access and engagement for neurodivergent applicants with extra time, adjusted fonts and a tailored user interface (UI) including background colours.
- ► Provide opportunities for candidates to familiarise themselves with the UI and format prior to their interview though a practice portal.
- ► Recommend generic, blank backgrounds for video or videoconference facilitated interviews or, if not possible, advise blurred backdrops.
- ► Ensure diversity of interviewers for prerecorded videos and those assessing them including gender, age and ethnicity, experts-by-experience, and other stakeholders.
- ► Avoid culturally sensitive subject areas, language, age and ability bias in interview content.
- ► Ensure the use of inclusive, gender-neutral language with appropriate pronouns for example, 'they/them/their'.

The focus of the article is to detail the study undertaken to evaluate the psychometric properties of the asynchronous MMI. We aimed to explore reliability (internal consistency) for all users with subgroup analysis by key characteristics (age, gender, nationality, disability/additional support needs), construct validity, dimensionality, acceptability and usability.

In this context disability refers to a person who self-identifies with physical or mental impairment, and the impairment has a substantial and long-term adverse effect on the person's ability to carry out normal day-to-day activities. Not all neurodivergent people consider themselves 'disabled' but instead neurological conditions are viewed a result of normal variations in the human genome but where additional support might be required. We, therefore, use the term 'support need' to include neurodivergent applicants.

## METHODS

The MMI questions were developed and tested by the university's community emulating our previously established in-person processes. A diverse range of individuals including academic staff, service users and practice partners were individually videorecorded asking MMI questions. The video recordings were uploaded and the system, configured in a seven question, 4 min MMI with 1 min between questions.

The asynchronous online MMI system was adopted for selection to Health Professions undergraduate programmes (Adult, Child, Mental Health Nursing, Midwifery and Paramedic Science) at one UK university during 2021/2022 recruitment cycle. All applicants to these programmes were invited to register for a 1 week

slot that was convenient for them. At the start of their selected slot, applicants were emailed a link to the system. Here they could access the practice portal to check their Wi-Fi speed, familiarise themselves with the UI/key functionality, become conversant with the process through a detailed instructional video, and practice a question. When they felt prepared, they recorded their interview answers. To mirror our previous in-person process, once they had begun, applicants could not stop their MMI and start again. Any technical issues including those without the necessary equipment and/or Wi-Fi connectivity were supported to enable their successful completion on an individual basis.

All individuals at the university who would usually interview applicants as part of their role took part in this new process. These included academic staff, service users and practice partners. They were invited to self-allocate to 1-week interview assessment slots. Thereafter, they received a link and code to access the system to review and assess applicant interviews against uploaded scoring rubrics live on the system. Each interviewer was allocated a number of applicant's video responses to a single question thereby aligning with the principles of an MMI. Also, emulating MMI methodology, a 'red flag' option could be ticked, and details populated in a text box if a cause for concern was raised in applicants' answers. Applicant's performance scores were downloaded at the end of the process to inform offer/reject decisions by university admissions officers.

University Admissions Officers conducted an internal resource evaluation to explore time and cost spent on the asynchronous MMI compared with face-to-face and Zoom-facilitated MMIs.

### Design

This study was underpinned by Olsen and Eoyang's theory of 'Complex Adaptive Systems'.[18] This is characterised by an adaptive and iterative working style appropriate when developing and optimising the system for fairness and reliability in the absence of any known precedent. We also grounded the system design in Gilliland's[19] justice-based model. According to their theory, a selection system's adherence to procedural and distributive justice rules promotes applicants' perceptions of fairness. Procedural justice rules relate to the approach used to derive decisions, in this case an asynchronous MMI. It includes the formal process (opportunity to perform and administration), explanation (feedback, information and transparency) and interpersonal communication throughout the selection process. Distributive justice rules encompass adherence to equity when determining selection outcomes.[19]

### Participant recruitment

All applicants (n=712 at the data collection point) to the UK university and all interviewers (n=96) who assessed applicants were invited to evaluate the system.

## Patient and public involvement

The university's service user group were supportive of the move to online asynchronous interviews. One service user acted as an interviewer by videorecording an MMI question. A further three were interview assessors, thereby continuing an established model at this university of service user involvement in recruitment. The service user group were also involved in the MMI question writing by reviewing draft questions and providing feedback. In the seven-question circuit, one practice partner video recorded an MMI question and eight assessed applicants' videos.

## Data collection

Applicant interview performance data, routinely collected to inform offer/reject decisions, were used in the reliability analyses. Applicant interviews were scored against 10 question-specific criteria on a seven-point Likert-type scale with descriptive anchors for the seven questions. Criteria details are withheld for test security. Applicant interview scores were summed for each applicant.

To assess usability, applicants were invited to complete an online semistructured evaluation questionnaire hosted on Qualtrics once they had received their interview outcome decision.

To assess acceptability, interviewers were invited to evaluate the process at the end of the recruitment cycle through an online semistructured questionnaire also hosted on Qualtrics.[20]

## Analysis

We used applicant scores across questions to explore reliability (internal consistency) for all users and a randomly selected subsample self-reporting key characteristics including gender, age (<20 or 20+), nationality (UK/non-UK) and disability/support needs (absence/presence) using Cronbach's alpha (Stata, V.16.1, StataCorp).[21] Cronbach's alpha, as a conventional measure of internal consistency (reliability), assumes that item responses are unidimensional, meaning that they are reflective of only one underlying construct or factor. However, this may not always be the case, therefore, scale dimensionality was assessed using a parallel analysis and confirmatory factor analysis (CFA).[22 23] The parallel analysis and reliability values were derived using the software package FACTOR.[24] The ordinal factor analyses were conducted in Mplus[25] V.8.8. Usability and acceptability were explored with descriptive statistics (closed questions) and conventional content analysis[26] (open questions).

## RESULTS

### Sample characteristics

Data were available from 712 applicants to nursing, midwifery and paramedic science programmes at the data collection point on 1 May 2022 and for key characteristics for the subsample (n=284) (table 1). Disabilities self-reported by interviewers were reduced hearing and visual acuity. Applicants reported neurodivergent challenges across the spectrum of dyslexia, dyspraxia and attention deficit hyperactivity disorder.

## Reliability and assessment of fairness

### Reliability

Applicant data were shown to be normally distributed (Kurtosis 0.000, p = 0.005) and symmetrical with skewness 1.0.

Internal consistency was good-excellent across scenario questions (for n=712 applicants) mean Cronbach's $\alpha$ 0.72 (range 0.64–0.89). Subgroup analyses showed similarly positive results with mean $\alpha$ female/male: 0.74/0.87 (range 0.70–0.89); age: <20 years/>21 years 0.76/0.83 (range 0.72–0.86), disability/additional challenges/non-disability: 0.88/0.78 (range 0.74–0.89) and UK/non-UK 0.78/0.0.77 (range 0.72–0.83) (table 2).

### Dimensionality/construct validity

The results of the parallel analysis suggested a maximum of seven dimensions underlay the response pattern. A Schmid-Leiman[27] transformation can be used to understand if a factor analytical model is best understood as a hierarchical in nature. This partitions observed variance into that explained by one or more general ('G') factors underlying three or more specific factors (a hierarchical model will be mathematically 'just identified' by three factors).[27] The results of the Schmid-Leiman transformed factor analysis indicated a seven-factor solution, relating to the MMI scenarios, with an underlying general factor which substantially loaded on all the former seven factors. An ordinal CFA to test this seven-factor hierarchical model showed an excellent fit to the data (Confirmatory Fit Index (CFI)=0.99, Tucker Lewis Index (TLI)=0.99, RMSE=0.034). In contrast, a one factor model showed a poor fit to the response data (CFI=0.70, TLI=0.69, RMSEA=0.19) (online supplemental appendix 2).

The results of the parallel analysis suggested a maximum number of seven plausible factors. The variance explained by additional postulated factors did not exceed that observed for the random data generated (online supplemental appendix 3).

### Usability (applicants)

The online evaluation was completed by 210 applicants (29% response rate). The majority were under 20 years of age, self-identified as white female and with representation from across programmes (table 3).

The majority of applicants had not undertaken an asynchronous MMI previously. Overall applicants found the instructions either helpful or very helpful and did not experience any technical issues. From a setup perspective, was considered the right amount of time by over half with one-third stating it was too short. We asked additional overarching open free text questions regarding applicants' views on online interviews (table 3) as well as their overall 'top' positives and 'top' negatives of system (table 4 summary, online supplemental table 1).

**Table 1** Participant self-identified characteristics

| Interview assessors n=65 | | | | Applicant subsample n=284 | | |
|---|---|---|---|---|---|---|
| | | N | % | | N | % |
| Age (years) | 25 and under | 0 | 0 | <20 | 193 | 68 |
| | 26–35 | 8 | 13 | | | |
| | 36–45 | 18 | 28 | >20 | 91 | 32 |
| | 45+ | 35 | 53 | | | |
| | Prefer not to say | 4 | 6 | | | |
| Gender | Female | 55 | 84 | Female | 230 | 81 |
| | Male | 7 | 10 | Male | 54 | 19 |
| | Other | 0 | 0 | Other | 0 | 0 |
| | Prefer not to say | 3 | 6 | Prefer not to say | 0 | 0 |
| Disability/ additional support needs | No | 60 | 93 | No | 228 | 80 |
| | Yes | 5 | 7 | Yes (64% neurodiverse, 34% mental health, 2% hearing challenges) | 58 | 20 |
| | Prefer not to say | 0 | 0 | Prefer not to say | 0 | 0 |
| Ethnicity | White (English/Welsh/Scottish/Northern Irish, Irish, Gypsy, Irish traveller, other white background) | 61 | 94 | UK/Ireland | 229 | 81 |
| | Mixed/multiple Ethnic Groups (white and black Caribbean, white and black African, any other mixed background) | 2 | 3 | Non-UK/Ireland | 55 | 19 |
| | Asian/Asian British (Indian, Pakistani, Bangladeshi, Chinese, any other Asian background) | 0 | 0 | N/A | N/A | |
| | Other (Arab, any other ethnic group) | 0 | 0 | | | |
| | Prefer not to say | 2 | 3 | | | |
| Role | University Health Sciences Staff | 46 | 71 | N/A | N/A | |
| | Practice partner | 10 | 16 | | | |
| | Service user | 9 | 13 | | | |
| | Prefer not to say | 0 | 0 | | | |

N/A, not available.

We received 158 separate positive comments and 140 negative comments in relation to the questions asking for 'top' positives and 'top' negatives, Notably, 93% of the positive comments centred around three themes: ease (40%), reduced stress (28%) and fair (25%). We received fewer negative comments overall and these were split into two main themes: limited direct communication (34%) and critique of the MMI process itself (32%). Nineteen per cent of respondents raised technical issues as a potential negative issue, however, these did not transpire for around 80% in reality. The 12 respondents who did experience technical issues cited: buffering (3), frozen screen (2), crash (2), video skipping (2), screen scaling (1), microphone (1) and upload (1). They all subsequently successfully completed their interview at a second attempt.

### Acceptability (Interviewers)
Sixty-five interviewers took part in the online evaluation representing a 71% response rate. The majority were white British female university staff, over the age of 45

years with no declared disability (table 1). This is representative of the University Faculty staff profile which is located in the Southeast of the UK.

Ninety-six per cent (n=62) of interviewers found the system intuitive, easy to use and reported a perceived reduction in stress. They primarily attributed this to increased convenience and flexibility. A 70%-time reduction was independently reported by our admissions officers. They estimated this based on time spent on other interview approaches (face-to-face MMIs and video-conference-facilitated MMIs) compared with our asynchronous MMIs; categorised into preinterview communications, setup, staff recruitment (including covering sick time), interview facilitation and postinterview communications (online supplemental appendix 4). The majority of the time saving was ascribed directly to the asynchronous modality which removed the need for staff to either facilitate face-to-face or online live interviews. Additionally, the asynchronous approach alleviated the pressures of last-minute non-availability of interviewers particularly

**Table 2** MMI question reliability (internal consistency) N=712

| Question (total mean) | Obs | Mean | SD | Min | Max | Item-test correlation | Item-rest correlation | Cronbach's alpha |
|---|---|---|---|---|---|---|---|---|
| 1 | 765 | 48.6732 | 10.3966 | 10 | 70 | 0.6541 | 0.4677 | 0.68 |
| 2 | 795 | 51.57107 | 9.70229 | 10 | 70 | 0.6434 | 0.4532 | 0.67 |
| 3 | 790 | 51.08101 | 9.10597 | 10 | 70 | 0.599 | 0.421 | 0.69 |
| 4 | 809 | 50.23486 | 8.97655 | 10 | 70 | 0.5826 | 0.4001 | 0.70 |
| 5 | 762 | 50.27428 | 9.88047 | 10 | 70 | 0.6624 | 0.4941 | 0.67 |
| 6 | 765 | 51.79869 | 8.7486 | 20 | 70 | 0.6132 | 0.4484 | 0.69 |
| 7 | 712 | 50.10499 | 11.0865 | 14 | 70 | 0.6076 | 0.3871 | 0.70 |
| Scale reliability coefficient | | | | | | | | 0.7216 |

**Subgroup reliability n=284**

| | Cronbach's alpha | | | | | | | |
|---|---|---|---|---|---|---|---|---|
| | Age | | Self-identified gender | | Self-declared disability | | Country of birth | |
| Question (total score) | <20 years | >20 years | Female | Male | Yes | Non | UK/Ireland | Non-UK/ Ireland |
| 1 | 0.74 | 0.83 | 0.73 | 0.87 | 0.88 | 0.76 | 0.78 | 0.74 |
| 2 | 0.74 | 0.83 | 0.72 | 0.87 | 0.88 | 0.76 | 0.77 | 0.75 |
| 3 | 0.75 | 0.83 | 0.74 | 0.87 | 0.87 | 0.77 | 0.78 | 0.77 |
| 4 | 0.77 | 0.85 | 0.76 | 0.89 | 0.88 | 0.79 | 0.79 | 0.79 |
| 5 | 0.73 | 0.82 | 0.71 | 0.86 | 0.87 | 0.75 | 0.76 | 0.73 |
| 6 | 0.77 | 0.84 | 0.76 | 0.87 | 0.87 | 0.79 | 0.79 | 0.79 |
| 7 | 0.79 | 0.87 | 0.78 | 0.88 | 0.89 | 0.80 | 0.80 | 0.86 |
| Mean | 0.76 | 0.83 | 0.74 | 0.87 | 0.88 | 0.78 | 0.78 | 0.77 |

MMI, multiple mini interview.

practice partners as they were not tied to one scheduled day/time but had a time period (1 week) within which they could assess the interview recordings.

None of the interviewer assessors stated that they had used an asynchronous online MMI previously. Almost all (96%) found it easy to use and the user interface intuitive (92%). Less than 10% reported technical issues other than download issues which were resolved. Five per cent were 'not accepting' of using the asynchronous MMI in the future.

We were interested to better understand whether interviewers felt communication could be assessed in an asynchronous modality. Thirty-three per cent said 'yes', 54% 'somewhat', while 13% of respondents (n=6) responded 'no'. To generate more in depth insights we asked interviewers their top positives and negatives of the system. These are presented in table 4. We received 132 positive and 90 negative comments. The majority of positive comments (94%) related to perceived convenience (62%), fairness (14%) ease of navigation (9%) and benefits for the applicant (9%). Negative comments were split more evenly into perceptions of their being less personal (30%), critique of MMI methodology (24%), limited communication assessment and ability to build rapport (23%), and 5% had technology-related process concerns. Six per cent cited 'none'.

## DISCUSSION

These findings suggest the online asynchronous MMI is reliable, fair, time-efficient and acceptable. The results of the factor analyses infer that there are scenario-level effects but that that these all relate to an underlying general factor indicative that the process is assessing different dimensions/constructs relevant to healthcare. These could be method effects or alternatively conceptualised as representing different aspects of the interpersonal procedural knowledge required to perform well on the MMI.

This platform is the only known custom built asynchronous online interview emulating the MMI methodology. Cognisant of Gilliland's[19] procedural and distributive justice rules, our aim was to optimise applicant accessibility through building principles for fairness into the MMI design and system setup. The reliability results and usability and acceptability evaluation signal this was largely achieved which is critical to applicants perception of fairness.[28] These data suggest the configuration of our asynchronous MMI resulted in an equitable process particularly with the familiarisation enabled through the practise portal.[15] We note the higher Cronbach's alpha for disability compared with non-disability. While this is reassuring, it merits further investigation with a larger sample size.

**Table 3** Applicant characteristics and usability evaluation

| N=210 | | | |
|---|---|---|---|
| Demographics | | N | % |
| Age | <20 | 120 | 57 |
| | 20+ | 90 | 43 |
| Self-identified gender | Female | 172 | 82 |
| | Male | 36 | 17 |
| | Other/non-binary | 2 | 1 |
| Nationality | UK/Ireland | 134 | 64 |
| | Non-UK/Ireland | 68 | 32 |
| | Prefer not to say | 8 | 4 |
| Programme representation | | | |
| Nursing | Adult | 42 | 20 |
| | Child | 34 | 16 |
| | Mental health | 52 | 25 |
| Midwifery | | 32 | 15 |
| Paramedic science | | 50 | 24 |
| Closed question responses summary data | | | |
| Had not taken an asynchronous MMI like this before | | | 94 |
| Found the instructions helpful/very helpful | | | 90 |
| Did not experience technical issues | | | 79 |
| Said the 1 min between questions was 'about right' | | | 75 |
| Found the probe questions helpful/very helpful | | | 66 |
| Four minutes was about the right amount of time | | | 65 |
| Open question free text responses | | | |
| Question | Response % | | |
| How do you feel about videorecording and uploading your MMI responses as part of a 'new look' interview process triggered by COVID-19 social distancing restrictions? | Happy/very happy | 67 | Not very happy/not at all happy  33 |
| Please tell us about your views on online interviews generally | Accepting/very accepting | 66 | Partly accepting/not at all accepting  35 |
| Do you see a future for online interviews like the one you have used with us? | Yes/definitely yes | 68 | Probably not/ definitely not  32 |

MMI, multiple mini interview.

Rice[29](p452) suggests that social presence or the 'degree to which a medium is perceived as conveying the presence of the communicating participants' impacts on applicant acceptability. Social presence plays a central role in trust, enjoyment and the perceived usefulness of the technological medium.[28] There was by definition an absence of actual social presence in the asynchronous modality. However, we sought to mitigate this through softening the intersection between human and technology by design. The majority said they found the system intuitive/very intuitive and simple to use. This infers that the UI design and inclusive language may have contributed towards a positive experience.

Applicants in this study (>66%) were either accepting or very accepting of the online asynchronous MMI with around one third (37%) agreeing we should 'definitely return' to face-to-face interviews.

Notably, only 2% (n=2) of applicants said they were 'not at all accepting' of the asynchronous MMI. The majority of applicants were 20 years or under. It could be suggested that a younger demographic are more familiar with and accepting of online technology and see it as part of their day-to-day lives. However, these data signal that there was no difference in reliability for those under 20 years compared with over 20 years old and that applicant performance was not impacted by their age.

Applicants' experiences, particularly perceptions of fairness, are of paramount concern for universities. The implications of fairness can extend to postinterview outcomes including offer/acceptance rates. It has been

**Table 4** Summary applicant and interviewer top positives and negatives

| **Applicant** | | | | |
|---|---|---|---|---|
| **Positives** | | | | |
| Theme | Sub theme detail | N=158 (%) | % of total comments | Illustrative quote |
| Ease | Access, intuitive, convenient, flexible, simple | 63 (40) | 21 | *'Quick, not time consuming, simple'* |
| Reduced stress | More relaxed in own home, take my time, breaks available, start when wish | 45 (28) | 15 | *'It's more relaxing to be in your own home instead of a new environment which for me is much less intimidating resulting in perhaps me performing better in the interview'.* |
| Fairer | Reduced costs (travel), reduced time away from other responsibilities (caring), the practice portal and availability of question text helped neurodiverse applicants | 39 (25) | 13 | *'It's much easier in regard to travel for those who live far away, don't have the funds or time or who have other commitments'.* *'Reduces the possibility of bias during the interview process'.* *'…I could do a practice question to get used to the layout of the interview, so I wasn't going into the interview completely blind'.* |
| 'Meet' staff | Able to see more of the university's community | 9 (7) | 3 | *'I liked that there were 7 different people asking the questions…'* |
| Covid safe | Travel not required | 2 (1) | 1 | *'…Has helped to continue the process of admissions in a positive way…'* |
| **Negatives** | | | | |
| Limited direct communication | Less personal, no conversation | 48 (34) | 16 | *'… Impersonal, I feel like I can't make a connection or read the interviewers body language in the interview…'* |
| MMI | Pressure felt due to the timed methodology, presence of the countdown timer | 45 (32) | 15 | *'4 minutes per question was quite pressuring to fill'.* *'Ticking down time was off-putting'.* |
| Technical issues | Wi-Fi cut out | 27 (19) | 9 | *'My screen froze, potential upload failure, loss of connection'.* |
| Don't get staff vibe | Not directly meeting staff | 2 (1) | 1 | *'Lack of interaction between the student-tutor, the dialogue'* |
| Cannot ask questions | | 18 (13) | 6 | *'Not personalised, not able to ask questions'* |
| **Interviewer** | | | | |
| **Positives** | | | | |
| Convenient | Can prioritise workload. Flexible. Quicker Less stressful | 82 (62%) | 37 | *'Efficient, less time consuming and able to prioritise your workload and complete interviews in chunks as opposed to one long stint'* |
| Fairer | Less unconscious bias Reduced travel (costs) | 18 (14%) | 7 | *'Reduced bias from different people asking the questions'.* *'Equity… no student gets help with prompts more than others'* |
| East to navigate | Simple | 12 (9%) | 5 | *'Easy to use with clear instructions'* |
| Benefits applicants | Less stressed in own environment | 12 (9%) | 6 | *'Separating the two events(interview/applicant day)will help manage anxiety and stress'* |
| Applicant assessment | Can get a (better) sense of the applicant | 4 (3%) | 2 | *'I feel I can get a sense of the applicant through this process'.* |

**Table 4** Continued

| Interviewer | | | | |
|---|---|---|---|---|
| Less worried about technical issues | Not one time/date dependent | 3 (2%) | 2 | 'Much less stressful as I don't have to worry about the internet connection'. |
| Ability to rewatch | If cause for concern' | 1 (1%) | 1 | 'Ability to rewatch for clarification'. |
| **Negatives** | | | | |
| Less personal | Don't get feel for applicant | 27 (30%) | 12 | 'I think it's better for applicants to speak to someone in person'. 'I question whether it produces the same quality of response'. |
| Critique of MMI methodology | Timed circuit/countdown timer | 22 (24%) | 10 | 'Encourages to me to make a snap decision'. 'Can't ask follow-on questions… permits no probing'. |
| Communication/ building rapport difficult | Less able to assess non-verbal communication, social skills, spontaneous cognitive ability. | 21 (23%) | 9 | 'Little in the way of holism'. 'Disconnect perhaps for applicants'. |
| Limited support for nervous applicants | Cannot help if upset or stressed | 8 (9%) | 4 | 'No opportunity to provide any support if the candidate appears upset. if they are struggling mentally or emotionally'. |
| Tech related process concerns | Tech issues Stress of Zoom/Teams online mode | 4 (5%) | 2 | 'Robotic process' 'Repetitive and boring watching multiple videos' 'Technical issues for some student may give them a disadvantage'. |
| Did not have any | Stated 'none' | 5 (6%) | 2 | 'Don't have any' |

MMI, multiple mini-interview.

suggested that applicants who perceive that recruitment and selection processes are fair are more attracted to organisations.[30–32] Concurring with Brenner et al[12] applicants reported 'perceived fairness' as one of their top three positives of the asynchronous interview system. Their reasons include reduced travel costs and time away from caring responsibilities, as well as enhanced familiarisation of the process through the practice portal. MMI interviews were prerecorded using inclusive language by diverse staff, representative of the university community. Additional time and an intuitive system UI appeared to help meet the needs of neurodiverse applicants.

Incorporating the 10 fairness principles was not difficult as many were low-cost design features that appeared to be impactful and generically applicable. We suggest these should become a default approach for online interviews used in health professional selection to enable applicant performance optimisation. In view of the paucity of published evidence,[15] these novel insights are informative as we inevitably move towards a technology-augmented future where asynchronous video interviews are considered a modality that is here to stay.[15]

We received a 60:40 ratio of positive to negative comments from interview assessors. The largest contributor (62%) to the overall feedback related to positivity around convenience including ability to prioritise workload, flexibility, speed and reduced stress. This was

followed by ease of navigation and reduced bias. These findings corroborate evidence garnered outside the field of healthcare where asynchronous interviews have been found to be faster, cheaper, require less employee time and open the applicant funnel to allow more people to be interviewed than would otherwise have had the opportunity.[15] Nevertheless, communication skills are central to the role of a health professional and are assessed as a generic skill/attribute in each MMI question at this university. One-sixth of interview assessors said they did not feel communication skills could be assessed while over one-third stated they could in the asynchronous modality. Further research is warranted to better understand the intersection between humans and technology including barriers and enablers to effective communication and communication assessment.

The largest contributing negative comment (30%) focused on a perception that the process was less personal. We might have anticipated this to be higher. Steps being considered to enhance personal connection include live chats and Q&A sessions as well as increasing the number of 'offer holder-days' provided by the university where applicants are invited onto campus to engage with staff without the stress of an interview clouding the experience. Further research evaluating the effectiveness of this strategy is suggested.

## Strengths and limitations of this study

Invitations to evaluate the system were sent out after applicants received notification of their interview outcome. We understand this may have impacted on their perception of the process however, we were required to adhere to the university's policy.

The study was a theoretically driven mixed-methods cross-discipline approach. However, we acknowledge the generalisability limitations of a single site design, but this was an essential step ahead of a planned large multisite international evaluation.

While the sample size is large, the low response rate from applicants is a potential limitation and may infer selection bias. Users' views could be impacted by many factors outside the scope of this research for example past experiences for which we were unable to account for.

It was not possible to conduct a comparison study 'pre/post system optimisation'. Covid necessitated a move to online interviews in unprecedented times. Data were not collected on applicant or interviewer views at the time given the burdens they were already facing. During that time, however, we explored how fairness could be optimised through a review of published literature with findings embedded in our system (online supplemental appendix 1). In a high stakes admission process, it would be ethically wrong to conduct a prospective study now to compare with and without the 10 principles for fairness given the apparent benefits.

Collapsing applicant data into UK/Ireland and Non-UK Ireland was a necessary pragmatic decision based on lack of consistent reporting of ethnicity between the university (who did not routinely retain applicant ethnicity data until enrolment) and the UK University Central Admissions System.

## Rigour

In spite of reassurances in all communications, we were mindful that applicants might be concerned that their evaluation could impact on their interview outcome hence the invitations were sent once offer/reject decisions had been communicated to applicants.

All data in this evaluation were independently analysed and peer reviewed by multiple authors (ACallwood, JH, SR and PAT). The qualitative content analysis was undertaken by two authors independeltly (AChristidis, SR). A <5% difference was noted between authors' findings. A compromise was mutually agreed in instances where this occurred.

## Conclusion

With 10 principles for fairness designed in, these findings suggest the asynchronous online interview is reliable, equitable, time-efficient and acceptable. It is a moral imperative that healthcare workforces represent the societies which they serve, however, unintended bias can influence selection decisions. In the absence of generically available consensus guidance on how fairness can be optimised in online interviews, these novel insights are applicable internationally across selection to health professions. Embedding fairness into the design of online interviews is relatively straightforward and low cost to implement. These data advance our understanding which is vital as we inevitably more towards a technology augmented future in the context of global workforce pressures.

**Acknowledgements** Grateful thanks to staff, students, practice partners and service users without whom this work would not be possible.

**Contributors** ACallwood is guarantor. ACallwood, LG and AChristidis contributed to the technical development. ACallwood, LG, AChristidis, JH, PAT, AK and SR contributed to the study design, data collection, drafting and revisions. ACallwood, JH, SR and PAT contributed to the data analysis. Author AChristidis is no longer directly affiliated to the university but was at the commencement of this study.

**Funding** This work was supported with funding from Innovate UK grant number 37043.

**Competing interests** ACallwood and LG are co-founders and AChristidis is an employee of Sammi-Select, funded by UKRI as a spinout company from the University of Surrey to build the platform.

**Patient and public involvement** Patients and/or the public were involved in the design, or conduct, or reporting, or dissemination plans of this research. Refer to the Methods section for further details.

**Patient consent for publication** Not applicable.

**Ethics approval** This study involves human participants and this study received a favourable ethical opinion (FEO) from the University Research Ethics committee (UEC/2017/111/FHMS). Participants gave informed consent where personal data was included

**Provenance and peer review** Not commissioned; externally peer reviewed.

**Data availability statement** Data are available on reasonable request.

**ORCID iD**
Alison Callwood http://orcid.org/0000-0001-9617-909X

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
