## [Reviewer comments · BMJ Open]

ARTICLE DETAILS

TITLE (PROVISIONAL)	Cross-sectional evaluation of an asynchronous Multiple Mini Interview (MMI) in selection to health professions training programmes with ten principles for fairness built-in
AUTHORS	Callwood, Alison; Harris, Jenny; Gillam, Lee; Roberts, Sarah; Kubacki, Angela; Christidis, Angelos; Tiffin, Paul

VERSION 1 – REVIEW

REVIEWER	Lillis, Steven University of Waikato, Student Health
REVIEW RETURNED	23-Apr-2023

GENERAL COMMENTS	The paper is of interest to training organisations as it describes a novel advancement of the MMI process. There are a number of issues with the paper that must be attended to. The major problem is its length. Many of the paragraphs in the introduction could be reduced to one or two sentences. The subject of equity should be addressed once and not repeated throughout the paper. The section on the rapid literature review does not need the level of data and outcomes that have been included. A feasibility analysis was undertaken by Admissions Officers. This is good as I am aware of several training organisations who dropped MMI for cost reasons. Yet, the only financial comment was that there was a substantial time saving. If you have further cost comparison data, please include it. Otherwise, I don't think you can claim that a feasibility analysis was really undertaken. The section on design starts off with 'This dual paradigmatic cross-sectional study...' but it is not made clear if the paradigms are that of complex adaptive systems and a justice based model or something else entirely. Again, there is a lot of redundant words. The section on factor analysis needs looking at. What was the purpose of undertaking a factor analysis? It appears to me that it revealed all questions were relevant and there was no redundancy of items. If so, this needs to be made clear. The level of detail on confirmatory factor analysis and the Schmid-Keiman analysis is excessive for the audience. This could be 'available if requested'. In the reliability analysis, it would be nice to have 'Alpha with item deleted' figures but this is a minor point. The 'Open question response' reporting: was it an open question of the response was a Likert scale? Why did you report on combined two responses (happy/very happy) as well as a single response (not very happy)? It is customary to collapse a 5 response scale to 3 responses; positive, neutral and negative. While you have described a lot of information in your report, a lot of it is institution specific and therefore of little relevance to other organisations.
--

	I note the comment “More broadly however, organisations should carefully consider how even the act of inviting an applicant to complete an asynchronous interview may impact justice perceptions and attitudes” This feels quite idealistic to the point of driving organisations to fearful indecision rather than useful advice. You need to think of your audience as being busy academics seeking solutions to problems. Was this solution valid (I think yes, from the construction of the process) cost effective, reliable and acceptable? Some of this is lost in the detail.
--	---

REVIEWER	Kumwenda, Ben University of Dundee, Centre for Medical Education
REVIEW RETURNED	02-May-2023

GENERAL COMMENTS	A good and timely article that I have read with a lot of interest. The authors have clearly explained the process for setting up the asynchronous MMI. I am not cognisant of Gilliland’s justice-based model and how applicable it is in this context. I will let other reviewers comment on that. However, I am satisfied with the trustworthiness and the quality of reporting of the psychometric properties of the asynchronous MMI. In terms of the analysis (page 11), there is a good sample size from the applicants. There is good coverage of face validity (both from the assessors’ perspective and the applicants’ perspective). It might be worth mentioning to the reader the reason why the analyst run Cronbach’s Alpha, ordinal factor analysis, and Dimensionality/Construct validity (Page 13, lines 34-58). A short statement will be helpful to the less statistics-savvy audience. I found the results in Table 2 a little interesting and I wonder if the authors noted that too. It appears that the sub-group reliability score (Cronbach’s alpha) of self-declared disability applicants is consistently stronger than that of applicants with no reported disability. Do the differences in inter-item correlation indicate that the reliability is stronger in one group than the other, or it is just a statistical noise that does not mean much in practice? Some of the summary breakdown presented in Table 3 is a duplication of what has been presented in Table 2 (age, gender) and does not offer much in terms of detail. Editing Table 3 will save you some space.
--

VERSION 1 – AUTHOR RESPONSE

viewer comments 1

1. *Cut down length overall.* Response: Word count reduced from 4826 to 4371
2. *Cut down introduction.* Response: Rapid review of literature detail moved to supplementary file to cut down content as well as additional refinement edits.
3. *Reduce number of times equity addressed.* Response: removed as able throughout.
4. *Rapid review of literature needs less detail.* Response: Rapid review of literature detail moved to supplementary file.
5. *Feasibility analysis undertaken by Admissions Officers limited to mention of timesaving. Either add detail or remove feasibility.* Response: The text states: ‘University Admissions Officers conducted an internal resource evaluation to explore time and cost spent on the asynchronous MMI compared with face-to-face and Zoom-facilitated MMIs. It does not specifically mention feasibility.

6. Clarify 'dual paradigmatic cross-sectional study'. Response: removed wording to avoid confusion.
7. Make the purpose and findings of the factor analysis clearer. Response: reworded.
8. Section on factor analysis needs reviewing with removal of CFA and Schmid-Keiman analysis to 'data available on request'. Response: request to retain for transparency and statistical rigour with additional details in Appendix 3.
9. Clarify 'open question reporting' ie origins as opposed to Likert scale. Response: free text added for clarity (to distinguish from the closed Likert scale responses).
10. Revise reporting of 'happy/very happy' and 'not very happy' so there is parity i.e. two combined responses for each not 1 for 'not very happy'. Response: Table 3 amended to include.
11. Explain relevance to other organisations in more detail. Response: additional text added e.g. pg 22 to demonstrate the generically applicable insights.
12. Revise comment 'more broadly organisations should carefully consider how even the act of inviting an applicant to complete an asynchronous interview may impact justice perceptions' ... Response: Deleted.

Reviewer 2

1. Explain further why Cronbach's alpha, ordinal factor analysis and dimensionality/construct validity were all run. Response: Explanation in the Analysis section.
2. Clarify sub-group analysis of self-declared disability vs no reported disability as appears stronger for self-declared disability. Response: Note added recognising this and suggestion that further analysis on a larger sample is recommended.
3. Double check demographics in Table 2 and 3 are not repeated. Response: these are two different data sets. Table 2 is the sub-group demographics from the random sub-sample. Table 3 are the applicant volunteers who took part in the evaluation.

VERSION 2 – REVIEW

REVIEWER	Lillis, Steven University of Waikato, Student Health
REVIEW RETURNED	08-Aug-2023

GENERAL COMMENTS	While noting the changes made by the authors, I still remain concerned about the length and density of this paper. The journal recommends 4,000 words as a maximum with occasional exceptions. The authors have reduced the paper from 4,800 to 4,300. It remains difficult to read even though I am familiar with the field. The central message is easily lost in the fine detail. For example, does the sentence "Moreover, evidence from recruitment to the police force demonstrates that fairness can also be optimised through incorporating language that supports the affirmation of values with increased probability of minority applicants passing an assessment by 50%" add anything to the message? There has not been an adequate response to the issue of feasibility. The paper states that cost comparison was made yet this data does not appear in the revision. This is important as increased cost is one of the reasons that this method of selection has been replaced in some institutions. Does Table 2 need to be in the main body? It may be better as an appendix or a single sentence in the body as there was remarkable consistency in the alpha across all items.
---

REVIEWER	Lillis, Steven
-----------------	----------------

	University of Waikato, Student Health
REVIEW RETURNED	08-Aug-2023

GENERAL COMMENTS	While noting the changes made by the authors, I still remain concerned about the length and density of this paper. The journal recommends 4,000 words as a maximum with occasional exceptions. The authors have reduced the paper from 4,800 to 4,300. It remains difficult to read even though I am familiar with the field. The central message is easily lost in the fine detail. For example, does the sentence "Moreover, evidence from recruitment to the police force demonstrates that fairness can also be optimised through incorporating language that supports the affirmation of values with increased probability of minority applicants passing an assessment by 50%" add anything to the message? There has not been an adequate response to the issue of feasibility. The paper states that cost comparison was made yet this data does not appear in the revision. This is important as increased cost is one of the reasons that this method of selection has been replaced in some institutions. Does Table 2 need to be in the main body? It may be better as an appendix or a single sentence in the body as there was remarkable consistency in the alpha across all items.
---

VERSION 2 – AUTHOR RESPONSE

Response to reviewers V28.08.23

Reviewer's comments 1: none suggested.

Reviewer's comments 2:

1. Reduce word count: Response - The word count has been reduced to 4003.
2. Additional detail needed regarding inclusion of cost-comparison data: Response - Example data has been included as an appendix (Appendix 2).
3. Feasibility not been adequately addressed: Response – we do not mention that this was a feasibility study.
4. Question whether Table 2 should be in the main body or appendix: Response - Suggest including it in main body as includes core findings.

VERSION 3 – REVIEW

REVIEWER	Lillis, Steven University of Waikato, Student Health
REVIEW RETURNED	16-Sep-2023

GENERAL COMMENTS	The required changes have been made, the paper now reads well.
--